# Electronegativity under Confinement

**DOI:** 10.3390/molecules26226924

**Published:** 2021-11-17

**Authors:** Andrés Robles-Navarro, Carlos Cárdenas, Patricio Fuentealba

**Affiliations:** 1Departamento de Física, Facultad de Ciencias, Universidad de Chile, Las Palmeras 3425, Ñuñoa, Santiago 7800003, Chile; andres.robles@ug.uchile.cl; 2Centro para el Desarrollo de la Nanociencia y la Nanotecnología (CEDENNA), Avda. Ecuador 3493, Estación Central, Santiago 9170124, Chile

**Keywords:** electronegativity, confined atoms, Hartree–Fock

## Abstract

The electronegativity concept was first formulated by Pauling in the first half of the 20th century to explain quantitatively the properties of chemical bonds between different types of atoms. Today, it is widely known that, in high-pressure regimes, the reactivity properties of atoms can change, and, thus, the bond patterns in molecules and solids are affected. In this work, we studied the effects of high pressure modeled by a confining potential on different definitions of electronegativity and, additionally, tested the accuracy of first-order perturbation theory in the context of density functional theory for confined atoms of the second row at the Hartree–Fock level. As expected, the electronegativity of atoms at high confinement is very different than that of their free counterparts since it depends on the electronic configuration of the atom, and, thus, its periodicity is modified at higher pressures.

## 1. Introduction

Electronegativity is one of the most important empirical concepts in chemistry, and the study of its variations in extreme conditions could be of importance. Confinement can simulate the effects of high pressure, of the order of gigapascals, which are pressures that can now be reached in the laboratory, and they are important in the study of phenomena occurring in the core of planets. Confinement can also simulate what happens to a system when it is placed inside a cavity like a zeolite or one from the fullerene family. Therefore, in this work, we first discuss some of the most important equations to calculate electronegativity, always keeping in mind that it is an empirical concept and so it is impossible to derive it from the laws of quantum mechanics. At some point of a theoretical development, one has to jump to empiricism.

The other point treated in this work is confinement, which can change the electronic structure of the atom. It can affect its reactivity and its possible catalytic properties, and, in solids, it can change, for instance, the crystallographic phase and induce superconductivity. One important case is the high-pressure electrides, for which a theoretical model has been recently presented [1]. We show the different forms to do the confinement, which can use rigid walls or soft walls. It can also be simulated using a cavity inside a dielectric medium. Then, we discuss our results for the atoms from hydrogen to neon.

### 1.1. Electronegativity

Electronegativity is one of the most powerful chemical concepts developed in the last century. Already twenty years after its formulation, it was taught as part of high school chemistry programs and is now an essential part of every general chemistry course. The concept of electronegativity was one of the many clever ideas developed by L. Pauling almost one hundred years ago [2]. He postulated it in his famous book and was defined as the capacity of an atom in a molecule to attract electrons. This definition is however not exempt of formal difficulties, starting with the question of what is meant by an atom in a molecule. It is not clear how to define an atom in a molecule, and, in fact, there are many different definitions. Therefore, until today, the best way to measure it has been discussed. Like most chemistry concepts, electronegativity is an empirical one. This means it cannot be derived from the physical laws of quantum mechanics. This is one of the principal reasons to have many different formulas to calculate it. Pauling gave a scale of electronegativity based on the formation enthalpy of diatomic molecules. His argument was to consider two atoms *A* and *B*; if both atoms have the same capacity to attract electrons, there is no polarization of the bond, and the AB molecule would have a perfect covalent bond. Additionally, its dissociation energy, $D_{AB}$, would be an average of $D_{AA}$ and $D_{BB}$. However, if one of them succeeds in attracting electrons, the bond would be polar, and the amount of polarity should be a measure of the electronegativity difference, $\Delta \chi_{AB}$. Hence, the final equation reads
(1)$$D_{AB}=\frac{D_{AA}+D_{BB}}{2}+\Delta \chi_{AB}^{2}$$

Fixing the electronegativity of the fluorine atom to the value of 4, he obtained a scale that works surprisingly well despite the fact that the units were already strange (eV${}^{-1/2}$). Very recently, Tantardini and Oganov published an improvement in the equation that gives the electronegativity as a dimensionless number [3].

However, today there is apparently a consensus that it should be an atomic property that can not depend on other atoms. There have recently been various well-written articles [4,5] giving a review of the concept. For the interest of this work, we briefly describe two of them. Allen [5] put forward what he called the spectroscopic electronegativity, which is defined as
(2)$$\chi =\frac{n_{s}I_{s}+n_{p}I_{p}}{n_{v}},$$
where $n_{l}$ and $I_{l}$ are the number of valence electrons in the *l* shell and the ionization energy of an *l*-electron, respectively, and $n_{v}$ is the total number of valence electrons. Using Koopmans’ theorem, this equation can be easily generalized as
(3)$$\chi_{V}=-\frac{1}{n_{v}}\sum_{i}n_{i}\varepsilon_{i},$$
where $\varepsilon_{i}$ is the orbital energy of the *i*th valence orbital, $n_{i}$ is the occupation number, and the minus sign is to ensure that the electronegativity is positive for free atoms. This equation will be recognized here as the average valence electron binding energy. Nonetheless, in the spirit of the original Allen’s formulation, another way to generalize Equation (Equation 2) is simply to take the absolute values of the orbital energies. Thus,
(4)$$\chi_{A}=\frac{1}{n_{v}}\sum_{i}n_{i}\left|\varepsilon_{i}\right|$$
will be called the Allen’s electronegativity here, and it is always a positive number. Both generalizations have the advantage of covering in the best way the open shell cases and the atoms with the *d* orbitals occupied. However, the slight difference between Equations (Equation 3) and (Equation 4) will prove to be significant in the case of confined atoms as shown below.

Another important definition of electronegativity was put forward by Mulliken [6] (we cited one of his last beautiful works, but his definition was done in 1934). The equation reads
(5)$$\chi_{M}=\frac{I+A}{2}$$
where *I* is the ionization potential, and *A* is the electron affinity of the atom. This equation has at least two important points. First, it is the only one that uses information about the ability of the free atom to obtain one more electron, which is measured by the electron affinity, and, second, this equation was elegantly derived by Parr et al. [7] in the context of density functional theory, giving this definition a solid theoretical ground. Using Koopmans’ theorem, one can approximate the ionization potential and the electron affinity with the energies of the highest occupied molecular orbital (HOMO ) and the lowest unoccupied molecular orbital (LUMO). Hence, one can calculate the electronegativity from one single calculation of the neutral system, avoiding the mix of different sources of error, especially in the calculation of anions.

### 1.2. Confinement

An atom or molecule can be confined in different ways. It can be confined using impenetrable walls, which means to change the Dirichlet boundary condition, which requires that the wave function goes to zero at some radius $R_{c}$. This approach has been successfully used by Garza et al. [8,9,10,11,12]. Another possibility is to use soft walls. It means to put some repulsive finite potential centered at $R_{c}$. This approach was extensively studied by Diercksen et al. [13,14,15,16,17] and others [18,19]. Another, more-recent approach is to use a model called the extreme pressure polarizable continuum model [20,21]. It consists in putting the atom inside an external continuum low-dielectric medium. All of them can be used to model the effects of pressure or the effects of encapsulating an atom in a cavity like a zeolite or a fullerene. They can also model a quantum dot. A pioneer in the topic is Connerade; he recently published a review [22] and many other interesting works [23,24].

The study of atoms or molecules under strong pressure is something very real. Today, it is possible to reach in the laboratory pressures of the order of hundreds of gigapascals, and, in the center of planets, it is believed the pressure can be of the order of terapascals. The physical and chemical properties of an atom under these extreme conditions can be very different from those of the free atom [18,25]. At a high enough pressure, the atoms can change their valence state [19]. It implies a change in the coordination number producing new solid phases. The reason can be found in the possibility of a crossing of different atomic energy levels. Hence, orbitals that are empty in the free atom can be occupied at high pressures. This effect has been experimentally detected [26,27,28]. An excellent review of these effects has the nice title “The Chemical Imagination at Work in very Tight Places” [29]. Another phenomenon occurs when the PV term (*P* is the pressure and *V* the volume) in the equation of state works against the binding of the electron and an induced autoionization occurs. Connerade has written an interesting review about it. In a different application, Cioslowski [30,31,32] has studied the effect of harmonic confinement in the harmonium atom. Interestingly enough, they found the emergence of Wigner molecules in three-dimensional Coulombic systems. Some years ago, Chattaraj et al. [33,34] began to study the effects of confinement on the chemical reactivity. They used different theoretical models to show that confinement has a significant effect on many classical chemical reactions. Very recently, Rahm et al. [35] did a very exhaustive study of the variation of electronegativity under pressure for the entire periodic table. In this work, we expanded the study to include the possibility of autoionization, and in that way to calculate the electronegativity, we used information of the LUMO avoiding the cases where the electronegativity of the anion is greater than that of the neutral atom.

In this work, we used a soft wall constructed with a harmonic potential centered at the nuclei. By varying the curvature of the potential, one can simulate the variations in pressure. Of course, one has to translate the curvature parameter to units of pressure. We did it in a previous work [18], but now we present the results in terms of the parameter ω, which will be defined in the next equation. Hence, the Hamiltonian to be used reads, in atomic units, as
(6)$$\hat{H}=-\frac{1}{2}\sum_{i}\nabla_{i}^{2}-\sum_{i}\frac{Z}{r_{i}}+\frac{1}{2}\sum_{j\neq i}\frac{1}{r_{ij}}+\frac{1}{2}\sum_{i}\omega^{2}r_{i}^{2},$$
where the first term is the kinetic energy operator, the second one is the nucleus–electron attraction, and the third term is the electron–electron repulsion. Those three terms represent the usual Hamiltonian of the free atom. The harmonic potential wall is represented by the fourth term, where the parameter ω gives the curvature of the potential, and, by changing its value, one can simulate changes in the pressure. We solved the Schrödinger equation using this Hamiltonian for the atoms of the second row, He to Ne.

As an extension of our work, we tested the Taylor series expansion of perturbation theory developed in the context of the density functional theory. Recognizing first that the electronic chemical potential, μ, is the negative of the electronegativity of Mulliken [36], we can calculate the changes in the electronic chemical potential under an external perturbation as
(7)$$d\mu =\int {\left[\frac{\delta \mu }{\delta v(r)}\right]}_{N}\delta v\left(r\right)dr=\int f\left(r\right)\delta v\left(r\right)dr,$$
where the functional derivative of the electronic chemical potential with respect to the external perturbation, δv(r), is taken at constant number of electrons, *N*. This functional derivative is just the definition of the Fukui function, f(r). For more details, see reference [37]. In our case, the external perturbation is the harmonic potential wall
(8)$$\delta v\left(r\right)=\frac{\omega^{2}}{2}\sum_{i}r_{i}^{2},$$
where the sum is over all the electrons. In the finite difference approximation, the Fukui function from above is
(9)$$f\left(r\right)=\rho_{N+1}\left(r\right)-\rho_{N}\left(r\right)$$

In this way, the working equation for the variations in the electronic chemical potential is
(10)$$d\mu =\frac{\omega^{2}}{2}\left({\langle r^{2}\rangle }_{N+1}-{\langle r^{2}\rangle }_{N}\right),$$
where, in the brackets, one has the mean value of $r^{2}$ for the atom with N+1 electrons, the anion, and for the neutral atom with *N* electrons. In the same way, it is easy to see that the variations of the total energy under the external perturbation of the harmonic potential is given by
(11)$$dE=\frac{\omega^{2}}{2}\langle r^{2}\rangle$$

The last two equations were to test the accuracy of the series expansion. The computational details are presented first, and then the results and discussion are presented.

## 2. Computational Details

The parameter ω controls the strength of confinement. The basis set for expanding atomic orbitals should include functions for the bound states of the Coulomb potential and functions for the states of the harmonic oscillator. For the part of the harmonic oscillator, we chose Gaussian functions with suitable exponents. Diercksen et al. [14,15,16,17] found that the optimal exponents follow the approximated series ω, ω/2, ω/4,⋯, ω/2n. We used the first four exponents of the series and included a basis set with angular momentum l = 0, 1, 2, 3. For the Coulombic part of the potential, we used a decontracted 6-311G(d,p) basis set. To have an idea of the size of the basis set of this scheme, in the case of fluorine, there were 67 basis functions. We expected the effect of the basis set to be more important in atoms with more electrons and open-shell configurations. All calculations were done at the Hartree–Fock level since in a previous work we demonstrated that the orbital energies are well represented at this level of theory [18]. Moreover, this work was more at the qualitative and empirical level. Gaussian09 software [38] was used to compute all the necessary integrals including the confining potential as an effective core potential operator centered in a ghost atom located at $r_{0}$. For the calculation of the electronegativity after Mulliken, Equation (Equation 5), we relied on Koopmans’ theorem and replaced the ionization potential by the energy of the HOMO and the electron affinity by the energy of the LUMO.

## 3. Results

### 3.1. Electronegativity Formulas under Confinement

To begin with, the ratio of the total atomic energy at confinement ω over the energy of the free atom is depicted in Figure 1. One can see that the energy diminishes in magnitude (becomes more positive) for all of them at any value of confinement within the range selected, with the hydrogen atom having the largest variation in its energy. Interestingly, already at low confinement, the energy of the lithium atom decreased more than the energy of the helium atom. This is so because the helium atom was already much more compact than lithium, where the valence electron in the orbital 2s was more compressed by confinement. The same occurred to the neon atom. At some point, there was a crossing of the energies of helium and beryllium, demonstrating that it is highly probable that, at enough confinement, the energy of the beryllium atom could be lower than that of helium, which is a sign that the periodic table of the elements could be different at high pressures.

Figure 2, Figure 3 and Figure 4 show the variation of the electronegativity at three distinct values of ω, (0,0.5,1.0), calculated in three different ways: (1) $\chi_{V}$ using Equation (Equation 3) in Figure 2; (2) $\chi_{A}$, Allen’s electronegativity through Equation (Equation 4), which is shown in Figure 3; and (3) the Mulliken formula for $\chi_{M}$, plotted in Figure 4. For the free atoms, ω=0, all curves were similar, and they failed at the noble gas atoms, giving them the highest values of electronegativity; besides that, the results agreed with the chemical knowledge. However, under confinement, the curves were more complicated and were against the chemical intuition. $\chi_{A}$ is by definition always positive and at ω=0.5 presents a minimum for the carbon atom; the fluorine atom had an electronegativity very similar to that of the lithium atom. At higher confinement, ω=1.0, the lithium atom presented a very high value and the electronegativity, instead of increasing and going to the right side of the periodic table of elements, diminished. The other two scales, $\chi_{V}$ and $\chi_{M}$, presented a qualitatively correct trend, a minimum at the lithium atom, and an increasing in their value until reaching the fluorine atom. However, they had the unpleasant feature of being negative, which has no meaning. The reason lies in the sign of the HOMO energy, which becomes positive at already low confinement, as can be seen in Table 1 for lithium and beryllium atoms.

In Figure 5, one can see the plot of the electronegativity after Mulliken versus the values of the parameter ω. It is clear that the formula fails completely at low ω, taking negative values in the lithium atom already at ω=0.1, and, for heavier atoms, such as fluorine, it becomes negative at a value of ω=0.5. The reason is that it is against the positive value of the energy of the HOMO.

### 3.2. Testing of the Perturbation Theory Equations

In this section, the validity of the equations of perturbation theory, (Equation 10) and (Equation 11), is tested. First, the variation in energy as the confinement parameter increases is plotted in Figure 6 for the Li, B, C, and F atoms. One can see that, at lower values of confinement, the changes in energy were very similar for all atoms. However, already at ω=0.3, the lithium atom’s curve separates from the other ones, and its change in energy was considerably smaller. One has to keep in mind that the total energy of Li was also much lower than that of the other atoms. Hence, in percentage terms, it may be that all changes in energy were similar.

Next, the variations in total energy for the same atoms is presented in Figure 7 as the quotient of $E_{\omega }/E_{0}$, $E_{\omega }$ being the energy at a confinement ω and $E_{0}$ the energy of the free atom. It is clear that the heavier the atom the better the agreement is. The larger deviations are in the lithium atom, as was indicated above, and the kink produced at ω=0.45−0.5 was due to the change in the ground state configuration where there was a quasidegeneration; a perturbation theory for degenerate states should be used [39]. The agreement between both curves for the fluorine atom until very high values of confinement was very good.

Now, we analyze the changes in electronegativity. We used the electronegativity of Mulliken because it corresponds to the negative electronic chemical potential, μ. Figure 8 shows the results for the atoms of Li, B, C, and F. The larger variations were for the lithium atom, and all the curves were almost a perfect straight line. This is so because the difference between the mean values of $\langle r^{2}\rangle$ for the anion and the neutral atoms followed $\omega^{-1}$, which then gave a linear dependence in the formula with respect to the confinement strength. In Figure 9, the plots of the total value of μ for the same atoms are shown. Here, the results using the perturbation series deviate already at low values of confinement, again giving better results for the heavier atoms.

In summary, the behavior of atoms at high confinement can be very different in comparison with the free atoms. As has been previously demonstrated, the periodic table of the elements looks very different and so does the reactivity of the different atoms. The calculation of the electronegativity seems complicated because of quasidegeneracies and the positive energy of the HOMO. At high pressures, it may be necessary to consider other parameters to measure the ability of an atom in a molecule to attract electrons.

## Figures and Tables

**Figure 1 molecules-26-06924-f001:**
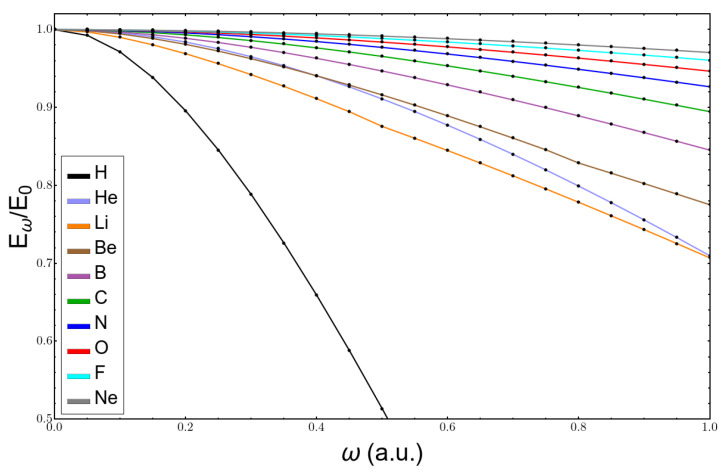
Variation of the ratio between the total energy of atoms from H to Ne at confinement ω and the total energy of their free counterpart with confinement.

**Figure 2 molecules-26-06924-f002:**
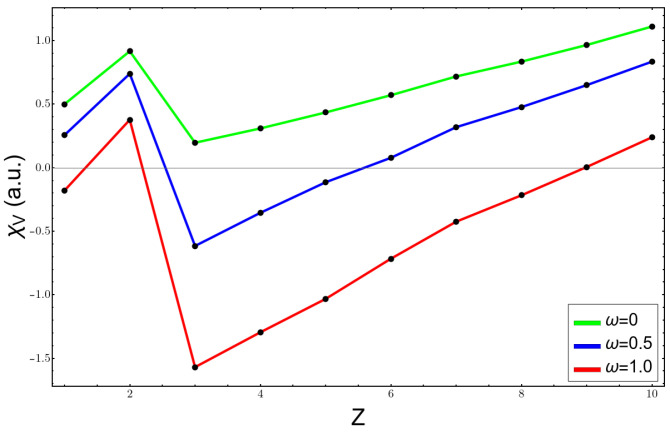
Electronegativity as the average electron binding energy at different confinement values for atoms from H to Ne.

**Figure 3 molecules-26-06924-f003:**
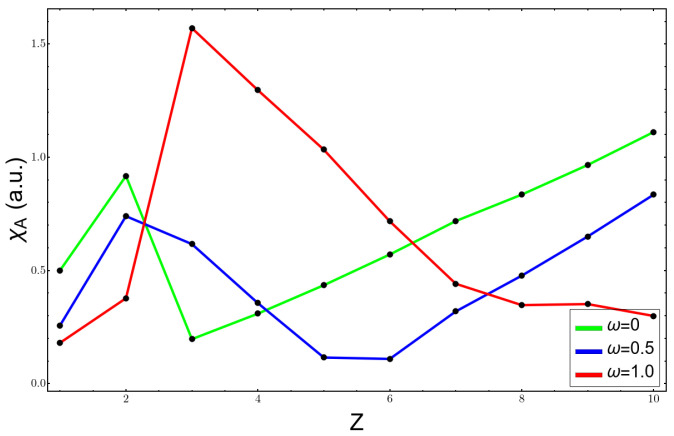
Allen’s electronegativity at different confinement values for the atoms from H to Ne.

**Figure 4 molecules-26-06924-f004:**
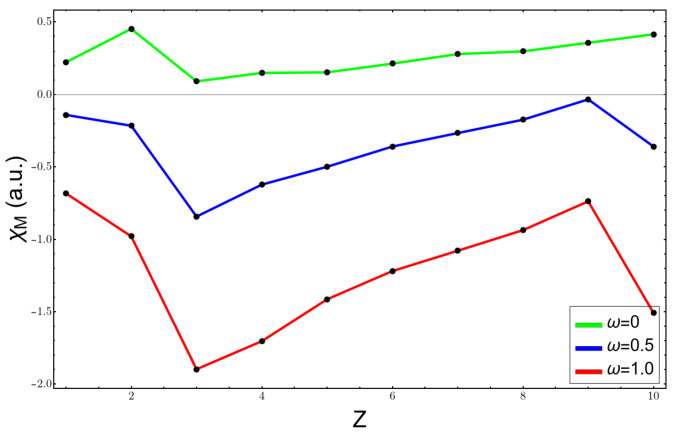
Mulliken’s electronegativity at different confinement values for the atoms from H to Ne.

**Figure 5 molecules-26-06924-f005:**
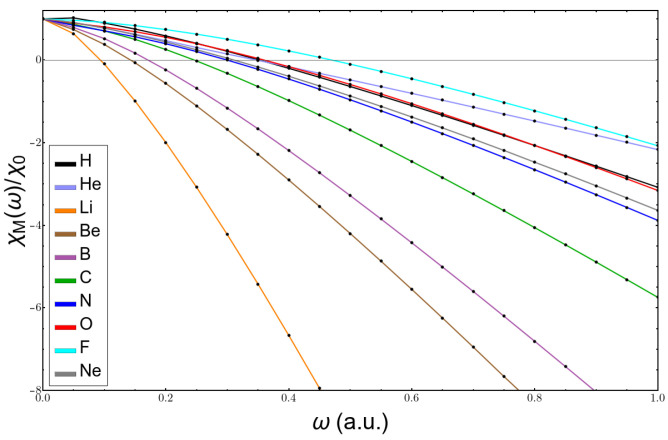
Variation with confinement of the ratio between Mulliken’s electronegativity as a function of ω and its free atom value.

**Figure 6 molecules-26-06924-f006:**
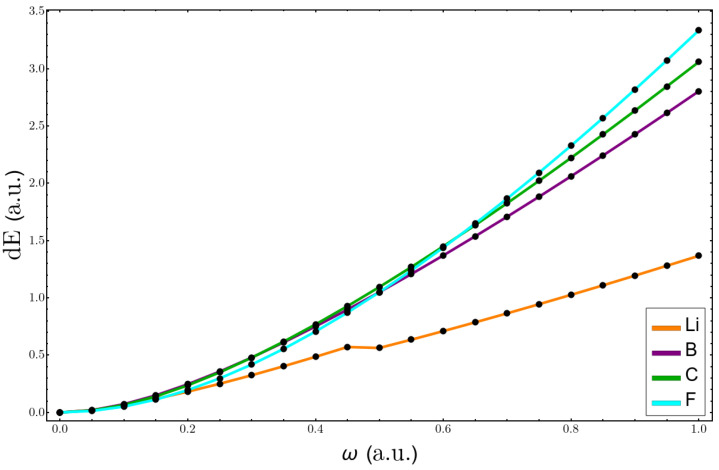
Change in the total energy with confinement, using Equation (Equation 11), for Li, B, C, and F atoms.

**Figure 7 molecules-26-06924-f007:**
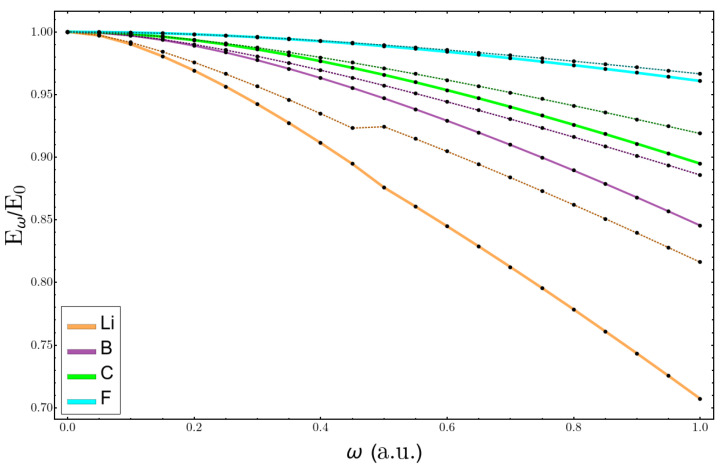
Change in the total energy with confinement relative to the unconfined atom. Solid lines correspond to the calculations solving the Hartree–Fock equations, and broken lines are the perturbation theory estimations.

**Figure 8 molecules-26-06924-f008:**
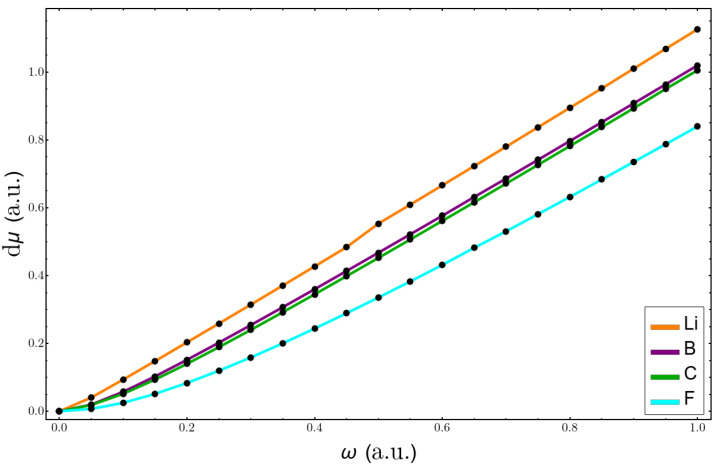
Change in electronegativity with confinement, using Equation (Equation 10), for Li, B, C, and F atoms.

**Figure 9 molecules-26-06924-f009:**
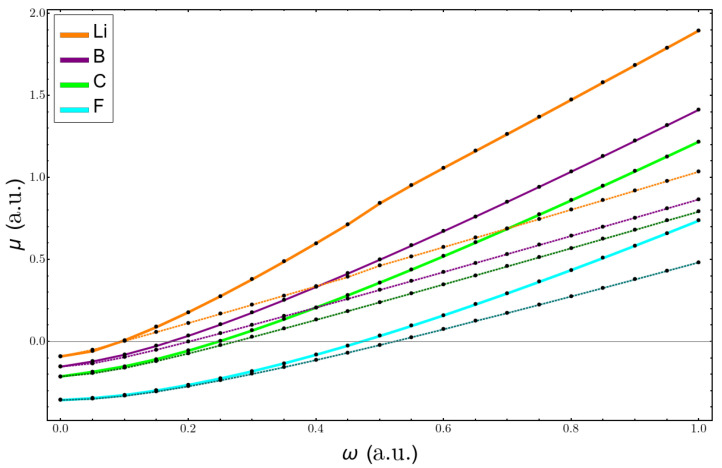
Variation in the electronic chemical potential with confinement. Solid lines correspond to the calculations solving the Hartree–Fock equations, and broken lines are the perturbation theory estimations.

**Table 1 molecules-26-06924-t001:** HOMO and LUMO energies (in atomic units) of the confined Li and Be atoms. Values in red correspond to the region in which a configuration crossing occurred.

ω	Li		Be	
	$\varepsilon_{\mathrm{HOMO}}$	$\varepsilon_{\mathrm{LUMO}}$	$\varepsilon_{\mathrm{HOMO}}$	$\varepsilon_{\mathrm{LUMO}}$
0.00	−0.19635	0.01645	−0.30897	0.01232
0.05	−0.17625	0.06057	−0.29534	0.07521
0.10	−0.12746	0.14397	−0.25898	0.14581
0.15	−0.06177	0.23963	−0.20751	0.22609
0.20	0.01528	0.34322	−0.14536	0.31262
0.25	0.10074	0.45273	−0.07509	0.40332
0.30	0.19280	0.56700	0.00166	0.49698
0.35	0.29026	0.68524	0.08380	0.59287
0.40	0.39225	0.80686	0.17054	0.69053
0.45	0.49813	0.93145	0.26127	0.78960
0.50	0.61623	1.07021	0.35554	0.88987
0.55	0.70756	1.20067	0.45297	0.99116
0.60	0.80017	1.31779	0.55325	1.09334
0.65	0.89375	1.43050	0.65612	1.19629
0.70	0.98821	1.54334	0.76137	1.29994
0.75	1.08350	1.65633	0.86880	1.40422
0.80	1.17953	1.76947	0.89158	1.58778
0.85	1.27685	1.88329	0.98810	1.71303
0.90	1.37364	1.99619	1.10064	1.83969
0.95	1.47179	2.10981	1.21471	1.96682
1.00	1.57025	2.22352	1.33044	2.07621

## Data Availability

The data is available on request from the author.

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
