# Peer review of "Electronegativity under Confinement"

_molecules, 2021, doi:10.3390/molecules26226924_

Round 1

Reviewer 1 Report

This work describes Hartree-Fock computations of single atoms inside spherical cavities. These computations are used for computing electronegativities, in different ways, as a function of confinement. The approach of comparing three different definitions of electronegativity within the same compression model is attractive. However, I have several major objections to this work, the primary one being that it, in the end, adds very little new insight. One approach followed is downright unphysical by construction, and another not properly considered. Proper consideration of past work and their conclusions is lacking throughout. The text, the language, also needs to be rewritten in a more precise manner, before being acceptable for publication.  

Currently there are plenty of sentences that are downright confusing. I will give only a few examples: “confinement can simulate the effects of high pressure..”. maybe this should read: “modeling of confinement can emulate the effect of high pressure”?

“In some point of a theoretical development one has to jump to empiricism”. Without becoming too philosophical here, what is meant? Is empirical data/experiments always necessary? With that, I disagree. Or are the authors trying to say what is rather obvious, that theory must relate to perceivable reality to be useful?

“The study of atoms or molecules under strong pressure is something very real.” One could argue that none has studied atoms under compression, nor molecules, but materials.

“It can affect its reactivity, its possible catalytic properties and…”

“Therefore, until today it is discussed the best way to measure it.”

There are way too many “it” in this manuscript, which makes IT hard to follow. Name the baby. A thorough comprehensive rewrite to clarify the text is necessary.   

The work is contradictory in places. For example, it (correctly!) states that quantities such as electronegativity cannot be derived from the physical laws of quantum mechanics. However, later in the work it is stated, relating to Mulliken’s definition of electronegativity that “..this equation was in an elegant way derived by Parr et al in the context of density functional theory, giving this definition a solid theoretical ground”. The definition of Parr is no more on solid theoretical ground than any other: All definitions of electronegativity are well defined. Many can be related to properties eminently computable using quantum mechanics. The fact that the conceptual DFT definition is popular and productively used is another matter entirely. This work muddles the waters when it comes to which definition is which: Mulliken is one. Another, arguably very similar, is commonly attributed to Parr (the negative of the chemical potential). However, that definition was really proposed by Iczkowski and Margrave, who are not cited at all.

It is confusing that the authors describe definitions within DFT, then directly jump to Koopman’s theorem, which famously does not hold for DFT. Only to jump back to calculations using Hartree-Fock.

One major flaw of this work begins with the assumption that there cannot be negative electronegativities. If you compress atoms in spherical potentials their electrons will eventually become less bound than a free electron in vacuum, and take positive energies. There is nothing unphysical about that. The electron is contained. The way this study has been conducted naturally leads to problems when implementing the Mulliken definition of electronegativity. The latter does not function well with negative electron affinities and ionization energies. Artem Oganov’s group have done this correctly, and made the Mulliken definition workable by considering the ionized electron dissolved in the compression medium. Also not cited.

The authors of this study then go on to “fix” a perceived issue with Lee Allen’s scale of electronegativity (sparsely cited). Instead of averaging over orbital energies, they considering the absolute value of these energies. Quite clearly, such an approach creates a completely artificial effect: when at some point during compression electrons should become unbound (with respect to vacuum) they are instead considered as bound (because the energy of orbitals are here always taken to be positive ). As a consequence, the main results (Figure 3) become unphysical. The work omits to mention the recent extension of Allen’s ideas by Rahm and Hoffmann under ambient conditions, who also address the challenges of defining the valence levels. This work also omits to mention that the “Allen definition”, have been extended to atoms at high pressure by those same authors.

There is a significant lack of comparison with well-established phenomena described in the literature. For example, the electronic configurational change in alkali atoms (and elements) have been discussed by several. Yet, the discussing on what is known previously, and citation to such work is sparse at best. Li is the only atom with such interesting effects in this study, but it is not even mentioned which states are involved, or why they occur.

The conclusion that “the behavior of atoms at high confinement can be very different in comparison with the free atoms” does, I am sorry to say, bring nothing new to the table. There are models of atomic compression over 70 years old showing this (none of the seminal work is cited). There is nothing wrong with being qualitative, but for this work to provide something new, it should at minimum compare relative trends, to other studies of the same systems. And reflect upon them. Are there meaningful difference in trends, and if so, are any of those important for our understanding of chemistry under compression?

Reviewer 2 Report

The manuscript sounds fine and with minor revision should be accepted. I have put my comments in the file PDF in attachment.

Reviewer 3 Report

In their paper Authors analyze the influnce of spatial confinement modeled by harmonic potential on the electronegativity of selected atoms. I find this work very interesting. It is a valuable addendum to the field of theoretical studies of the spatial confinement phenomenon.

I have two minor comments:

1) The English language and style is, in my opinion, in some places not appropriate for scentific publication.  Therefore, moderate changes in style would be required.

2) Authors should mention in the Introduction section how spatial confinement, especially in the form of harmonic potential, affects other properties of atoms and molecules, e.g. the nonlinear optical properties (see for example: PCCP, 2017, 19(11), pp. 7568–7575, PCCP 2017, 19(35), pp. 24276–24283).

Reviewer 4 Report

I enjoyed reading this paper . Chemistry under extreme conditions ( for example pressure ) is a hot and relevant topic and the work presented by the authors is situated in this field exploring the influence of pressure on atomic electronegativities. So it is timely and moreover ,it is very well suited for this Linus Pauling commemorative issue : as nicely explained  in the Introduction Pauling's definition of electronegativity has influenced generations of chemist's and , although in the paper other approaches are used , the link with Pauling is very well established . Although the literarture on confinement and pressure is getting more and more extended (as adequately shown in the authors' reference list ) the approach which is proposed is in my view  a refreshing  one in the sense that an analytical solution is presented to the confinement problem for the properties of   many-electron atoms by using an harmonic potential leading to a computable expression for the change in chemical potential ( or minus the electronegativity in the Mulliken ansatz). Eqn 11 is simple and attractive . The results and discussion section is easy to follow , where the issue of sign inversion of the orbital energies  , autoionization ,negative electron affinities and changes in electronic configurations are appropriately addressed. Figures and tables are well designed.

In summary , I find this an excellent contribution to the Special Linus Pauling issue of Molecules and I recommend publication as it stands .